# CM$^2$: Cross-Modal Contextual Modeling for Audio-Visual Speech Enhancement

## Abstract

Audio-Visual Speech Enhancement (AVSE) aims to improve speech quality in noisy environments by utilizing synchronized audio and visual cues. In real-world scenarios, noise is often non-stationary, interfering with speech signals at varying intensities over time. Despite these fluctuations, humans can discern and understand masked spoken words as if they were clear. This capability stems from the auditory system's ability to perceptually reconstruct interrupted speech using visual cues and semantic context in noisy environments, a process known as phonemic restoration. Inspired by this phenomenon, we propose Cross-Modal Contextual Modeling (CM$^2$), integrating contextual information across different modalities and levels to enhance speech quality. Specifically, we target two types of contextual information: semantic-level context and signal-level context. Semantic-level context enables the model to infer missing or corrupted content by leveraging semantic consistency across segments. Signal-level context further explores coherence within the signals developed from the semantic consistency. Additionally, we particularly highlight the role of visual appearance in modeling the frequency-domain characteristics of speech, aiming to further refine and enrich the expression of these contexts. Guided by this understanding, we introduce a Semantic Context Module (SeCM) at the very beginning of our framework to capture the initial semantic contextual information from both audio and visual modalities. Next, we propose a Signal Context Module (SiCM) to obtain signal-level contextual information from both raw noisy audio signal and the previously acquired audio-visual semantic-level context. Building on this rich contextual information, we finally introduce a Cross-Context Fusion Module (CCFM) to facilitate fine-grained context fusion across different modalities and types of contexts for further speech enhancement process. Comprehensive evaluations across various datasets demonstrate that our method significantly outperforms current state-of-the-art approaches, particularly in low signal-to-noise ratio (SNR) environments.

## 1 Introduction

Speech Enhancement (SE) aims to improve the intelligibility and quality of speech by eliminating background noise. This task has numerous applications in challenging acoustic environments. Traditional audio-only speech enhancement methods (AOSE) exclusively utilizes auditory signals, and struggles in scenarios with very low SNR (Defossez et al., 2020; Thakker et al., 2022a; Cao et al., 2022). Insights from cognitive psychology, particularly phenomena like the "Cocktail Party Effect" (Cherry, 1953) and the "McGurk Effect" (McGurk & MacDonald, 1976), highlight the importance of incorporating visual cues into speech enhancement. These phenomena have led to the flourishing development of Audio-Visual Speech Enhancement (AVSE), which integrates both auditory and visual modalities (Pan et al., 2021; Yang et al., 2022; Wang et al., 2023; Wu et al., 2024).

In real-world scenarios, background noises, including traffic, natural, and indoor noises, are predominantly non-stationary. This variability in noise distribution causes substantial interference in certain speech segments, while others are comparatively less affected.

Research in neuroscience, psychology, and phonetics offers crucial insights into addressing this issue. The human auditory system has the capability to reconstruct speech segments obscured by noise through semantic and signal contexts, a process termed *phonemic restoration* (Sunami et al., 2013; Sivonen et al., 2006). Inspired by this phenomenon, we propose that two types of contextual

Figure 1: **The overview of our work.** Two types of contexts—semantic and signal contexts—can help the human auditory system's phonemic restoration process. **Semantic Context:** Fluctuating noise levels in speech allow for varying degrees of semantic information retrieval. In segments with less interference, semantic information is clearer, which could provide reference to its neighboring noisy segments. **Signal Context:** Frames with heavy distortion or occlusion can often be interpreted based on adjacent, less affected frames. **Visual Frequency:** Visual attributes of a speaker, like gender and body shape, correlate closely with the audio frequency characteristic. For instance, gender and body shape suggest that females typically have higher pitches than males (Female-1 vs. Male-1), and heavier individuals higher than thinner ones (Male-2 vs. Male-1).

information—semantic and signal—are necessary to significantly enhance the quality and intelligibility of disrupted speech segments. Semantic context leverages continuity at the semantic level of speech to infer disrupted content, while signal context utilizes correlations at the frame level to aid in enhancement.

Subsequent research (Abbott & Shahin, 2018) indicates that visual cues significantly bolster *phonemic restoration*. Human visual features correlate strongly with audio characteristics like timbre and pitch (Kim et al., 2019). For instance, gender and body shape suggest that females typically have higher pitches than males, and heavier individuals higher than thinner ones. Despite this, most existing AVSE methods (Iuzzolino & Koishida, 2020b; Gao & Grauman, 2021; Wang et al., 2023) focus solely on temporal alignment during audio-visual modality fusion, neglecting the correlation between visual cues and audio frequencies.

Building on these insights, we propose Cross-Modal Contextual Modeling (CM$^2$), integrating contextual information across different modalities and levels to enhance speech quality. Our contributions could be summarized in three folds:

1. Given the challenge of non-stationary noise in speech enhancement, and drawing inspiration from the operational principles of the human auditory system, we have introduced two complementary forms of contextual information to support AVSE: semantic and signal context. By extracting and integrating these types of context across different modalities and domains, our approach significantly enhances the quality of speech.

2. We highlight and have experimentally validated the critical role of visual information along the audio frequency domain. This discovery holds potential to inspire future research across the audio-visual community.

3. Comprehensive evaluations on four composite datasets clearly show the advantages of our work. Especially, our approach consistently outperforms the current state-of-the-art (SOTA) methods across a wide range of SNR levels. Notably, at a signal-to-noise ratio (SNR) of -15 dB, our method achieves substantial relative improvements, enhancing the Signal-to-Distortion Ratio (SDR) by 63.6%, the Perceptual Evaluation of Speech Quality (PESQ) by 58.1%, and the Short-Time Objective Intelligibility (STOI) by 20.3%, relative to existing benchmarks. Significant enhancements continue even at an SNR of 0 dB, improving by 24.5% in SDR, 44.4% in PESQ, and 5.6% in STOI.

## 2 RELATED WORK

**Audio-Only Speech Enhancement:** AOSE has long been the predominant method for speech enhancement and remains a critical foundation in the field of speech signal processing. Initially, AOSE relies on statistical priors of noise data, employing techniques such as spectral subtraction (Boll, 1979), Wiener filtering (Lim & Oppenheim, 1978), and the minimum mean square error method Ephraim & Malah (1984). Subsequently, AOSE transitions to data-driven deep learning approaches, surmounting the limitations of traditional methods in handling non-white noise. AOSE methods are broadly categorized into two types: Time domain (T-domain) (Fu et al., 2018; Pandey & Wang,

2019; Defossez et al., 2020; Thakker et al., 2022b) and Time-Frequency domain (TF-domain) methods (Lu et al., 2013; Kolbæk et al., 2017; Fu et al., 2019; Cao et al., 2022). While T-domain methods typically estimate the audio waveform directly, TF-domain approaches can directly estimate the spectrum (Fu et al., 2017; Strake et al., 2020) or compute the spectrum by predicting a mask (Wang & Wang, 2013; Williamson et al., 2015; Fu et al., 2019; Cao et al., 2022). Our work is also based on the TF-domain mask approach, but we distinguish from AOSE by using supplementary visual cues.

**Audio-Visual Speech Enhancement:** Inspired by these researches in cognitive psychology (Cherry, 1953; McGurk & MacDonald, 1976), methods that introduce visual information for speech enhancement have emerged (Fisher III et al., 2000; Smaragdis & Casey, 2003; Parekh et al., 2017). Numerous AVSE approaches (Gabbay et al., 2018b; Hou et al., 2018; Afouras et al., 2019; Michelsanti et al., 2019; Pan et al., 2021; Gao & Grauman, 2021; Wang et al., 2023; Wu et al., 2024) have made efforts in enhancing modal fusion to maximize the benefits of both modalities. Iuzzolino & Koishida (2020a) introduced cross-modal squeeze-excitation mechanism for audio-visual fusion, which outperforms single channel-wise cancatenated fusion strategy. Wang et al. (2023) facilitated robust dynamic fusion of audio and visual modalities by assessing the dynamic reliability of each modality. Li et al. (2024a;b) implemented top-down attention for audio-visual fusion at multi temporal scales, mimicking the audio-visual pathways in the brain. Given the inherent synchronization of audio and visual modalities along the temporal dimension, most existing approaches focus solely on the fusion in the temporal domain and overlook the potential correlations between the frequency dimensions of the visual and audio modalities. In contrast, our work highlight the role of visual features in enhance the frequency dimension of audio.

**Phonemic Restoration:** Phonemic restoration is a phenomenon identified in cognitive psychology and neuroscience where listeners can perceptually "fill in" missing sounds in a speech signal, using contextual cues from the surrounding auditory and linguistic information (Warren, 1970; Samuel, 1981; Bashford et al., 1992; Riecke et al., 2008; Shahin et al., 2009; Powers & Hevey, 2016). Beyond linguistic contexts, visual cues are also proved to take part in facilitating the brain's phonemic restoration(Abbott & Shahin, 2018). This phenomenon has been extensively studied to understand both its neural underpinnings and its applications in improving communication aids for the hearing impaired (Shahin et al., 2009; Powers & Hevey, 2016). The cognitive mechanisms behind phonemic restoration is linked to top-down processing, where the brain utilizes contextual information and linguistic knowledge to reconstruct missing speech sounds (Samuel, 1981). This is particularly evident in conditions where speech is interrupted with silent intervals or noise bursts (Warren, 1970). These phenomena have inspired us to investigate the role of contexts in speech enhancement.

# 3 OUR PROPOSED CM$^2$

The goal of CM$^2$ is to integrate semantic and signal contexts across modalities for AVSE. We integrate our semantic context and signal context into a simple GAN-based model, consisting of a Generator and a Discriminator, as illustrated in Figure 2.

**Inputs Formulation:** Let $A \in \mathbb{R}^{1 \times T_a}$ and $V \in \mathbb{R}^{H \times W \times T_v}$ represent the noisy speech waveform and video frames, respectively, where $T_a$ denotes the length of the noisy signal; $H, W$, and $T_v$ denote the height, width, and the number of video frames, respectively. For the noisy speech $A$, a short-time Fourier transform (STFT) converts the waveform into a complex spectrogram $X \in \mathbb{R}^{2 \times T_x \times F_x}$, where $T_x$ and $F_x$ denote the time and frequency dimensions, respectively. Subsequently, the three distinct components of $X$ are concatenated along channel dimension to form the spectral input $X' \in \mathbb{R}^{3 \times T_x \times F_x}$ as:

$$X' = \langle X_m, X_r, X_i \rangle, \tag{1}$$

where $X_m$, $X_r$, and $X_i$ represent the magnitude, real, and imaginary components of the spectrogram, respectively, and $\langle \cdot, \cdot \rangle$ denotes the concatenation operation. We denote the target clean speech as $s$, and its corresponding complex spectrum as $S$.

**Generator:** Initially, the raw noisy speech $A$ and corresponding video $V$ are taken as input to a Semantic Context Module (SeCM), which extracts cross-modal semantic contexts $E \in \mathbb{R}^{B \times C_e \times T_e}$, where $B, C_e$ and $T_e$ denote batch size, channels and temporal dimensions, respectively. Concurrently, the spectrograms $X'$ are mapped through an audio encoder into high-dimensional audio features $P \in \mathbb{R}^{B \times C \times T_x \times F_x'}$, where $C$ denotes channels and $F_x'$ denotes frequency dimensions. Then, a Signal Context Module (SiCM) is employed to extract preliminary signal contexts $I \in \mathbb{R}^{BF_x' \times T_x \times C}$

Figure 2: The overall pipeline of $CM^2$. $CM^2$ consists of 8 main components: audio encoder, Semantic Context Module (SeCM), Signal Context Module (SiCM), Cross-Context Fusion Module (CCFM), Time-Frequency Block (TFBlock), magnitude decoder, complex decoder, and Discriminator. The notation $\times N$ in dashed box denotes that the same block repeats $N$ times.

based on the raw audio features $P$. Subsequent stages take the cross-modal semantic contexts $E$, TF-domain audio features $P$, and preliminary signal contexts $I$ as inputs to the Cross-Context Fusion Module (CCFM), which would further modeling the signal contexts. After this integration, $N$ Time-Frequency Blocks based on SiCM are employed to jointly model the time and frequency dimensions. The pipeline culminates with two decoders that output a complex spectrogram $\hat{S}$ and a magnitude spectrogram $\hat{S_m}$. These spectrograms are finally combined to produce the estimated clean speech.

**Discriminator:** The discriminator aims to estimate the non-differentiable key metric, PESQ, enabling it to serve as a training objective. It takes as input both the clean magnitude spectrum $S_m$ and the enhanced spectrum $\hat{S}_m$. During training, the output is the estimated PESQ score of the enhanced speech, with the generator's objective being to optimize the score of the enhanced speech towards 1.

In the subsequent sections, we will sequentially introduce each component of generator.

## 3.1 SEMANTIC CONTEXT MODULE

The Semantic Context Module (SeCM) is designed to extract semantic contexts from the audio-visual input , laying the foundation for subsequently enhancing target speech through semantic-level continuity. Inspired by the success of speech recognition models (Martinez et al., 2020) in extracting features closely linked to semantic information, we introduce three manners to construct our SeCM.

**SeCM$_V$ (Visual Speech Recognition-based Module):** In this manner, we introduce a simple learnable visual module similar to VSR models to obtain the initial semantic contexts. Specifically, we adopt the common structure cascaded by a 3D convolutional layer, a ResNet18 backbone, and a four-layer Temporal Convolutional Network (TCN), which will be learned from scratch for the target AVSE task with the input of grayscale facial frames $V \in \mathbb{R}^{H \times W \times T_v}$.

**SeCM$_{PV}$ (Pre-trained Visual Branch):** In the second manner, we take the pre-trained large-scale Audio-Visual speech representation models to obtain rich semantic contextual information. Specifically, models like AVHuBERT (Shi et al., 2022a), VATLM (Zhu et al., 2024), or Auto-AVSR (Ma et al., 2023) can all serve as our SeCM. Given that these models were originally pre-trained to obtain the shared semantic information between paired clean audio and video data, the corrupted noisy audio input in our task would lead to a discrepancy with the synchronized video data in these models. Therefore, we take only the visual modality as input to these models to obtain developed semantic information. The SeCM$_{PV}$ remains frozen during our training phase with the input of gray-scale frames of the mouth Region of Interest (ROI) following Shi et al. (2022a); Zhu et al. (2024).

**SeCM$_{PAV}$ (Pre-trained Audio-Visual Module):** For the third option, we introduce models that were pre-trained with noisy speech and paired videos, enabling them to more effectively handle noisy audio inputs and efficiently extract common semantic information from both noisy audio speech and video. Specifically, we adopt the robust AVHuBERT (Shi et al., 2022b) here to obtain robust semantic information. SeCM$PAV$ shares the same visual input as SeCM$PV$, with the addition of noisy speech waveforms $A$ also provided as input.

In summary, each variant of the SeCMs output the semantic context information at different degrees, collectively denoted as $E \in \mathbb{R}^{C_e \times T_e}$, where $C_e$ and $T_e$ denote channel and temporal dimensions respectively. The semantic context is subsequently used for signal context modeling and cross-context fusion in the later stages.

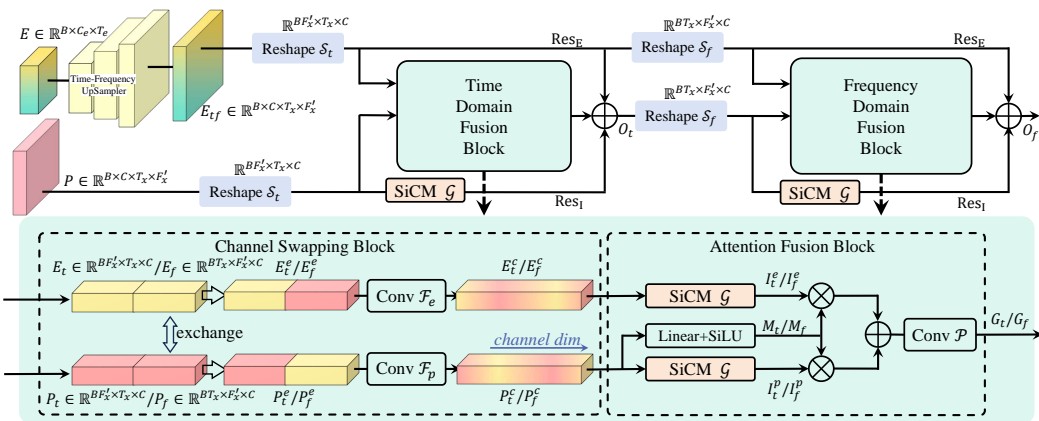

Figure 3: Structure of our CCFM. The CCFM consists of a time-frequency upsampler, a time-domain fusion block, and a frequency-domain fusion block. Both the time-domain and frequency-domain fusion blocks share an identical structure, which includes a channel swapping block and an attention-based fusion block. The only difference lies in the ordering of the input feature dimensions.

### 3.2 AUDIO ENCODER

The audio encoder aims to map the noisy spectral input $X'$ into a high-dimensional feature space, enabling subsequent context extraction and modeling. The encoder consists of 4 convolution blocks. The first convolution block comprises a convolution layer, an instance normalization (Ulyanov, 2016) and a PReLU activation (He et al., 2015), being used to extend the three input features to an intermediate feature map with $C$ channels. The middle two convolution blocks each include one convolution layer, a BatchNorm, and a residual connection. The final convolution block mirrors the structure of the first block and is tasked with reducing the frequency dimension to $F_x'$ to decrease complexity. It then outputs the high-dimensional audio feature $P \in \mathbb{R}^{C \times T_x \times F_x'}$.

### 3.3 SIGNAL CONTEXT MODULE

The Signal Context Module (SiCM) aims to extract signal contexts from both the raw noisy audio input features and the previous semantic contexts. The SiCM's target requires it possesses the capability to capture patterns inherent in continuous sequences. To this end, we employ a two-layer Bidirectional Mamba module (Bimamba) (Gu & Dao, 2023) for the global sequential-level modeling together with a convolutional module to enhance local modeling capabilities. Specifically, assuming the input sequence for signal context modeling is $Q_{in}$, two separate Mamba modules are introduced to process the original version, $Q_{in}$, and its reversed version, $Q_{in}^{(r)} = \mathcal{R}(Q)$, outputting $Q_{out}$ and $Q_{out}^{(r)}$ respectively. By reversing $Q_{out}^{(r)}$ again to transform it into the normal order, it is added to $Q_{out}$ to obtain the final contextual feature $Q_{OUT}$. The contextual feature incorporates residual connections, formulated as $Q_{OUT} = Q_{OUT} + Q_{in}$ $\mathcal{R}(\cdot)$ denotes the reverse operation. By modeling the sequence in both directions simultaneously, $Q_{OUT}$ can be considered to fully express the patterns inherent in the entire sequence. Finally, we employ a convolutional module $\mathcal{F}$ to $Q_{OUT}$ to enhance the local fine-grained information to produce the final output $Q_o$. The convolutional module also includes a residual connection.

$$Q_{out} = \text{mamba}(Q_{in}), \; Q_{out}^r = \text{mamba}(\mathcal{R}(Q_{in})), \tag{2}$$

$$Q_{OUT} = Q_{out} + \mathcal{R}(Q_{out}^r) + Q_{in}, \tag{3}$$

$$Q_o = \mathcal{F}(Q_{OUT}) + Q_{OUT} \quad Q_{in}, Q_{out}, Q_{OUT}, Q_o \in \mathbb{R}^{B,L,C}, \tag{4}$$

where $B, L,$ and $C$ denote the batch size, length, channels of features, respectively. The convolutional module is composed of several key components: it begins with a layer normalization, followed by a pointwise convolution layer, and a Gated Linear Unit (GLU). This is then succeeded by a depthwise convolution layer, integrated with a Swish activation function. The structure is further refined by an additional pointwise convolution layer, culminating in a dropout layer to regulate overfitting.

### 3.4 CROSS-CONTEXT FUSION MODULE

The target of $CM_2$ is to effectively aggregates the rich multi-modal cross-context information to guide the speech enhancement process. We introduce Cross-Context Fusion Module (CCFM) to

perform fine-grained contexts fusion across modalities and dimensions. CCFM receives inputs comprising the previous semantic contexts $E \in \mathbb{R}^{B \times C_e \times T_e}$ and the TF-domain audio features $P \in \mathbb{R}^{B \times C \times T_x \times F'_x}$ produced by the audio encoder. As illustrated in Figure 3, CCFM consists of a Time-Frequency Upsampler (TF-Upsampler), a time domain fusion block and a frequency do-main fusion block. The time-domain and frequency-domain fusion blocks share the same structure, each consisting of a channel swapping block and an attention-based fusion block. All the inputs are reshaped into a 1-d sequence before being fed into a fusion blocks or a SiCM , allowing for model-ing along specific dimensions. For instance, before being input into the Time Domain Fusion Block, the audio features $P \in \mathbb{R}^{B \times C \times T_x \times F'_x}$ are reshaped into the form $\mathbb{R}^{BF \times T \times C}$, where the frequency dimension $F$ is folded into the batch size. The reshaping operations preceding the time-domain and frequency-domain Fusion blocks are denoted as $\mathcal{S}_t$ and $\mathcal{S}_f$, respectively.

**Time-Frequency Upsampler:** A speaker's visual characteristics can provide valuable insights into their vocal attributes within the frequency domain, as detailed in Section 1. Thus, our approach incorporates considerations of both time and frequency dimensions, diverging diverging from tra-ditional AVSE methodologies that predominantly focus on temporal alignment. Specifically, our time-frequency upsampler expands both the time and frequency dimensions of the semantic con-texts $E$ to produce $E_{tf}$ for subsequent context fusion. This upsample module is equipped with two blocks, each consists of a transconvolution, a BatchNorm2d, and a PReLU activation function. The process can be summarized as:

$$E \in \mathbb{R}^{B \times C_e \times T_e \times 1} \xrightarrow{upsample} E_{tf} \in \mathbb{R}^{B \times C \times T_x \times F'_x}, \tag{5}$$

**Channel Swapping Block (CSBlock):** Given the distinct emphasis of channel information within the two modalities, channels that are redundant in one modality can provide complementary benefits to the other. Based on this idea, we introduce a Channel Swapping Block (CSBlock) to preliminarily merge information across the modalities at the channel level for later fine-grained fusion on time or frequency dimensions. Specifically, for the Time Domain Fusion Block, the input semantic context $E_t f$ and audio features $P$ are first reshaped to 1d sequence as $\mathcal{S}_t(E_{tf}) \rightarrow E_t \in \mathbb{R}^{BF'_x \times T_x \times C}$ and $\mathcal{S}_t(P) \rightarrow P_t \in \mathbb{R}^{BF'_x \times T_x \times C}$. Next, the second half of channels from $E_t$ and $P_t$ are exchanged to produce $E_t^e$ and $P_t^e$, respectively. Subsequently, $E_t^e$ and $P_t^e$ are processed together through 1d convolutions to effectively blend their channel information. We can denote the CSBlock as:

$$E_t^e = \langle E_t[...,:C//2], P_t[...,C//2:] \rangle, \tag{6}$$

$$P_t^e = \langle P_t[...,:C//2], E_t[...,C//2:] \rangle, \tag{7}$$

$$E_t^c = \mathcal{F}_e(E_t^e), \ P_t^c = \mathcal{F}_p(P_t^e). \tag{8}$$

**Attention-based Fusion Block (AFBlock):** After the CSBlock, the output contain both high-level semantic information and fine-grained sequential information from the raw noisy speech. Building on this fusion, we introduce SiCM $\mathcal{G}$ to extract the enhanced signal contexts $I_t^p$ and $I_t^e$ from $P_t^c$ and $E_t^c$, respectively. Then, a linear layer followed by a SiLU activation layer is applied on $P_t^c$ to generate the attention map $M_t \in \mathbb{R}^{BF'_x \times T_x \times C}$. Given that $P_t^c$ and $E_t^c$ exhibit symmetrical struc-tures, the selection of either feature for generating the attention maps does not influence subsequent outcomes. In this instance, we choose to generate $M_t$ from the features $P_t^c$. Subsequently, $M_t$ is element-wise multiplied with $I_P$ and $I_E$ and fused through addition. Finally, a 2D convolution $\mathcal{P}$ is applied to the attention result, forming the fused cross-modal signal contexts $G$. The whole process can be summarized as:

$$M_t = \text{SiLU}(\text{Linear}(P_t^c)), \ M_t \in \mathbb{R}^{BF'_x \times T_x \times C}, \tag{9}$$

$$I_t^e = \mathcal{G}(E_t^c), \ I_t^p = \mathcal{G}(P_t^c) \tag{10}$$

$$G_t = \mathcal{P}(I_t^e \odot M_t + I_t^p \odot M_t), \ G_t \in \mathbb{R}^{BF'_x \times T_x \times C}, \tag{11}$$

where $\odot$ denotes element-wise multiplication.

Additionally, the semantic-level and signal-level contexts are connected via residual connections to the output of the Fusion Block to form the cross-modal cross-context output $O_t$:

$$O_t = E_t + G_t + \mathcal{G}(P_t). \tag{12}$$

For the frequency domain fusion block, the operation mirrors that of the time domain fusion block. It is important to note that due to the sequential structure we design, the input audio features $P_f$ for frequency domain fusion block are derived from the output of the previous time domain fusion block, denoted as $O_t$. The final output of the CCFM is $O_f$. The process can be denoted as:

$$P_f = \mathcal{S}_f(O_t) \in \mathbb{R}^{BT_x \times F'_x \times C}, \tag{13}$$

$$O_f = E_f + G_f + \mathcal{G}(P_f). \tag{14}$$

## 3.5 Time-Frequency Block

Our Time-Frequency Block (TFBlock) includes two sequentially connected SiCM, each dedicated to extracting signal context from the time domain and frequency domain of the enhanced feature sequence post-fusion. Specifically, the features output by the CCFM or the previous block also require reshaping to transform them into a format where the target dimension is represented as a 1D sequence (e.g. $\mathbb{R}^{BF'_x \times T_x \times C}$). The process of TFBlock can be summarized as:

$$O_{i+1} = \mathcal{G}(\mathcal{S}_f(\mathcal{G}(\mathcal{S}_t(O_i)))), \ i \in \{1, 2, ..., N\}, \tag{15}$$

where $N$ denotes the number of TFBlocks, and the initial input features $O_1$ are given by $O_f$.

## 3.6 Decoders

Similar to prior methods in time-frequency domain speech enhancement (Cao et al., 2022), we utilize two decoders to independently predict the Ideal Ratio Mask (IRM) of the magnitude spectrum, as well as the real and imaginary components of the complex spectrum. These two decoders share the same architectural framework as the encoder, with the exception of two minor adjustments. Firstly, the downsampling operation applied to the frequency dimension in the encoder has been replaced by an upsampling operation, aimed at restoring the frequency dimension of the spectrum. Secondly, the magnitude decoder outputs with a single channel, while the complex decoder features two output channels, aligning with their respective target outputs.

Specifically, the magnitude decoder generates the Ideal Ratio Mask (IRM) of the spectrum, which is subsequently applied via point-wise multiplication to the input magnitude spectrum. Then, the complex decoder directly predicts the real and imaginary components $(\hat{S}'_r, \hat{S}'_i)$ of the complex spectrum. Following Yu et al. (2022); Li et al. (2022), the final complex spectrogram is obtained by the combination of the estimated magnitude $\hat{S}_m$ and the noisy phase $X_p$:

$$\hat{S}_r = \hat{S}_m \cos(X_p) + \hat{S}'_r \quad \hat{S}_i = \hat{S}_m \sin(X_p) + \hat{S}'_i \tag{16}$$

Finally, an inverse short-time Fourier transform (ISTFT) is applied to get the enhanced speech $\hat{x}$.

## 3.7 Loss Functions

For the learning process, we employ mean squared error (MSE) as the loss function on magnitude spectrogram $\mathcal{L}_m$ and complex spectrogram $\mathcal{L}_{ri}$:

$$\mathcal{L}_m = \text{MSE}(X_m, \hat{X}_m), \tag{17}$$

$$\mathcal{L}_{ri} = \text{MSE}(X_r, \hat{X}_r) + \text{MSE}(X_i, \hat{X}_i). \tag{18}$$

For the adversarial training, the generator loss $\mathcal{L}_{gan}$ and discriminator loss $\mathcal{L}_d$ can be expressed as

$$\mathcal{L}_{gan} = \text{MSE}(Dis(S_m, \hat{S}_m), 1), \tag{19}$$

$$\mathcal{L}_d = \text{MSE}(Dis(S_m, S_m), 1) + \text{MSE}(Dis(S_m, \hat{S}_m), PESQ_{gt}), \tag{20}$$

where $Dis(\cdot, \cdot)$ refers the discrminator and $PESQ_{gt}$ refers to the normalized PESQ score. The final optimization loss are employed as follows:

$$\mathcal{L}_G = \alpha \mathcal{L}_m + \beta \mathcal{L}_{ri} + \gamma \mathcal{L}_{gan}, \tag{21}$$

where $\alpha, \beta$, and $\gamma$ are the weights of the corresponding losses, chosen to achieve equal importance.

## 4 Experiments

### 4.1 Experimental Set up

#### 4.1.1 Datasets

We performed evaluation on the same set as most prior AVSE studies (Gao & Grauman, 2021; Xu et al., 2022; 2023; Wang et al., 2023). These datasets include LRS3 (Afouras et al., 2018) paired with DNS4 (Dubey et al., 2022), GRID (Cooke et al., 2006) paired with CHiME3 (Barker et al., 2015), TCD-TIMIT (Harte & Gillen, 2015) paired with NTCD-TIMIT (Abdelaziz et al., 2017), and MEAD (Wang et al., 2020a) paired with DEMAND (Thiemann et al.). For each of the above dataset pairs, the first dataset supplies paired clean audio and video, while the second dataset provides the noise. The training and test data are constructed by adding randomly selected noise samples into the clean data, as most established methods done.

Due to space constraints, we only present the experimental results on the widely-used LRS3 + DNS4 dataset here. For detailed results of other datasets, please refer to the Appendix.

### 4.1.2 IMPLEMENTATION DETAILS

In all experiments, audio samples are resampled to a sampling rate of 16 kHz, and the frame rate for videos is set to 25 fps. The length of speech segments is consistently set at 2 seconds. The STFT/ISTFT is performed using a Hamming window of 400 units in length and a hop size of 100 units. The frame rate of videos is set to 25fps. For facial inputs of $SeCM_V$, the dimensions are set to $H = W = 112$; for lip inputs of $SeCM_{PV}$ and $SeCM_{PAV}$, $H = W = 88$. During training, random cropping and horizontal flipping are introduced. In testing, only center cropping is utilized. For training, the noise SNR ranges from -15 to 0 dB. The loss weights are empirically established as $\alpha = 0.9$, $\beta = 0.1$, and $\gamma = 0.05$. The number of TFBlocks is set to $N = 4$.

### 4.2 RESULTS

#### 4.2.1 COMPARISON WITH STATE-OF-THE-ART METHODS

We compare the proposed CM$^2$ with other state-of-the-art (SOTA) models in Table 1. The results clearly show that our method significantly outperforms the current SOTA methods across all SNR ranges and evaluation metrics. Specifically, CM$^2$ achieves substantial average improvements in key metrics: PESQ sees a 53.6% enhancement, SDR improves by 39.1%, and STOI gains 12.075%, highlighting its effectiveness in speech enhancement across noisy conditions. Particularly under low signal-to-noise ratio conditions, our model demonstrates a more significant performance improvement. For example, at an SNR of -15 dB, our method improves SDR by 63.6%, the PESQ by 58.1%, and the STOI by 20.3%, relative to existing benchmarks. Our enhancement results at -15 dB even surpass those achieved by the DualAVSE method at 0 dB.

| Model | -15dB | | | -10dB | | | -5dB | | | 0dB | | |
|---|---|---|---|---|---|---|---|---|---|---|---|---|
| | SDR | PESQ | STOI | SDR | PESQ | STOI | SDR | PESQ | STOI | SDR | PESQ | STOI |
| DEMUCS 2020 | 2.33 | 1.210 | 0.561 | 5.84 | 1.297 | 0.682 | 9.10 | 1.443 | 0.777 | 11.85 | 1.631 | 0.839 |
| AV-DEMUCS 2020 | 3.03 | 1.213 | 0.611 | 6.15 | 1.314 | 0.694 | 9.47 | 1.483 | 0.787 | 11.86 | 1.666 | 0.843 |
| MuSE 2021 | -1.02 | 1.160 | 0.568 | 2.82 | 1.230 | 0.648 | 5.97 | 1.320 | 0.731 | 8.53 | 1.460 | 0.797 |
| VisualVoice 2021 | 2.52 | 1.317 | 0.643 | 5.73 | 1.475 | 0.735 | 8.16 | 1.682 | 0.808 | 10.32 | 1.963 | 0.865 |
| DualAVSE 2023 | _4.45_ | _1.435_ | _0.700_ | _7.54_ | _1.643_ | _0.780_ | _9.96_ | _1.909_ | _0.843_ | _12.32_ | _2.241_ | _0.889_ |
| **CM$^2$** | **7.28** | **2.269** | **0.842** | **10.39** | **2.608** | **0.886** | **13.00** | **2.922** | **0.917** | **15.34** | **3.235** | **0.939** |

Table 1: Comparisons with SOTA AVSE methods. All metric values are better when higher. **Bold** indicates the optimum results. Underlining indicates the suboptimum results.

#### 4.2.2 ABLATION STUDY

**Semantic Context (SeC):** To ascertain the efficacy of semantic context, we conducted evaluations on various configurations of SeCM as outlined in Section 3.1.

Firstly, we sequentially evaluated the effectiveness of the three manners as described in Section 3.1. For $SeCM_{PV}$, we utilize a version that was pre-trained on the clean LRS3 and VoxCeleb2 (Chung et al., 2018) datasets. For $SeCM_{PAV}$, we take both the noisy audio and the synchronized video data as input. As shown in Table 2, all SeCM variants significantly boost model performance, regardless of the specific manner of SeCM implemented. Notably, even the $SeCM_V$, developed from scratch, markedly improves speech quality under the challenging condition of -15 dB SNR. Furthermore, when noisy audio data is incorporated during the pre-training phase, $SeCM_PAV$ exhibits enhanced capabilities, with SDR and PESQ scores showing an average improvement of 10% over the baseline. The results confirm our asseration that semantic context is crucial for the AVSE task.

To conduct a more detailed assessment of how various degrees of semantic information affect the AVSE task, we utilize features from different encoder layers of robust AV-HuBERT as examples for evaluation. As indicated in Table 3, features from mid-to-high layers generally outperform those from lower layers. Typically, performance improves with higher-layer levels. The results supports again our motivation of introducing semantic context for AVSE because of the consensus that features extracted from higher layers are more closely associated with semantic-level cognition. Nevertheless, there are also exceptions; for example, under -5dB and 0dB SNR ratios, features from the 12th layer yield a higher SDR than those from the 24th layer. Given that different metrics assess

| SeC | -15dB | | | -10dB | | | -5dB | | | 0dB | | |
|---|---|---|---|---|---|---|---|---|---|---|---|---|
| | SDR | PESQ | STOI | SDR | PESQ | STOI | SDR | PESQ | STOI | SDR | PESQ | STOI |
| - | 5.171 | 1.904 | 0.752 | 8.509 | 2.241 | 0.831 | 11.393 | 2.616 | 0.884 | 14.005 | 2.992 | 0.921 |
| $SeCM_V$ | 5.414 | 1.973 | 0.779 | 8.637 | 2.307 | 0.845 | 11.526 | 2.663 | 0.891 | 14.210 | 3.022 | 0.924 |
| $SeCM_{PV}$ | 6.112 | 2.183 | 0.826 | 9.344 | 2.503 | 0.872 | 12.178 | 2.829 | 0.907 | 14.725 | 3.143 | 0.932 |
| $SeCM_{PAV}$ | **6.680** | **2.214** | **0.833** | **9.838** | **2.548** | **0.879** | **12.588** | **2.865** | **0.912** | **15.054** | **3.174** | **0.935** |

Table 2: Evaluation of SeC. '-' denotes that no semantic contexts are used (AOSE Baseline).

different aspects of speech quality, these findings reveal that semantic contexts from different layers prioritize distinct aspects of speech. This suggests that integrating features across various layers could potentially enhance the overall performance for AVSE.

| Layer | -15dB | | | -10dB | | | -5dB | | | 0dB | | |
|---|---|---|---|---|---|---|---|---|---|---|---|---|
| | SDR | PESQ | STOI | SDR | PESQ | STOI | SDR | PESQ | STOI | SDR | PESQ | STOI |
| $24^{th}$ Layer | **7.284** | **2.269** | **0.842** | 10.385 | **2.608** | **0.886** | 12.998 | **2.922** | **0.917** | 15.339 | **3.235** | **0.939** |
| $23^{rd}$ Layer | 7.257 | 2.246 | 0.840 | 10.339 | 2.589 | 0.885 | 12.838 | 2.897 | 0.916 | 15.003 | 3.207 | 0.938 |
| $12^{th}$ Layer | 7.140 | 2.164 | 0.826 | **10.351** | 2.511 | 0.878 | **13.049** | 2.858 | 0.913 | **15.439** | 3.189 | 0.937 |
| $1^{st}$ Layer | 5.814 | 2.096 | 0.805 | 9.475 | 2.405 | 0.863 | 12.377 | 2.751 | 0.903 | 14.858 | 3.085 | 0.930 |
| - | 5.171 | 1.904 | 0.752 | 8.509 | 2.241 | 0.831 | 11.393 | 2.616 | 0.884 | 14.005 | 2.992 | 0.921 |

Table 3: Evaluation of semantic contexts extracted from different encoder layers of robust AV-HuBERT. '-' denotes that no semantic contexts are utilized (AOSE Baseline).

**Signal Context (SiC):** To assess the importance of the SiC for AVSE, we conducted comparisons against two different choices: the Bimamba-based SiCM and the conformer-based version. Both Bimamba and Conformer possess sequence modeling capabilities to obtain signal context, but Bimamba demonstrates superior performance. The results presented in Table 4 demonstrate that stronger signal context leads to significantly improved performance in both AOSE and AVSE. Especially, within AVSE, the signal context contributes to a significant performance enhancement, with SDR showing an average improvement of 0.71 dB in AVSE, higher than the improvement of 0.69 dB in AOSE. This further supports our motivation that robust signal context is crucial for AVSE.

| Fusion Strategies | -15dB | | | -10dB | | | -5dB | | | 0dB | | |
|---|---|---|---|---|---|---|---|---|---|---|---|---|
| | SDR | PESQ | STOI | SDR | PESQ | STOI | SDR | PESQ | STOI | SDR | PESQ | STOI |
| Conformer | 4.311 | 1.780 | 0.730 | 7.797 | 2.138 | 0.817 | 10.742 | 2.496 | 0.875 | 13.467 | 2.875 | 0.914 |
| SiCM (ours) | **5.171** | **1.904** | **0.752** | **8.509** | **2.241** | **0.831** | **11.393** | **2.616** | **0.884** | **14.005** | **2.992** | **0.921** |

(a) Evaluation of SiCM and Conformer Modules in AOSE.

| Fusion Strategies | -15dB | | | -10dB | | | -5dB | | | 0dB | | |
|---|---|---|---|---|---|---|---|---|---|---|---|---|
| | SDR | PESQ | STOI | SDR | PESQ | STOI | SDR | PESQ | STOI | SDR | PESQ | STOI |
| Conformer | 4.656 | 1.845 | 0.755 | 7.952 | 2.161 | 0.829 | 10.831 | 2.521 | 0.88 | 13.492 | 2.897 | 0.917 |
| SiCM (ours) | **5.414** | **1.973** | **0.779** | **8.637** | **2.307** | **0.845** | **11.526** | **2.663** | **0.891** | **14.210** | **3.022** | **0.924** |

(b) Evaluation of SiCM and Conformer Modules in AVSE.

Table 4: Evaluation of SiC. For the experiments in Table(b), the visual inputs are processed by $SeCM_V$. The context fusion strategy is a simple additional fusion.

**Cross-Context Fusion Module (CCFM):** To demonstrate the efficacy of our CCFM, we compare the performance of two fusion strategies: using our proposed CCFM for context fusion and employing a simple addition approach. In the addition-based fusion, we removed the CCFM module from $CM^2$, but retained the time-frequency upsampler to ensure alignment between semantic contexts and raw audio features. Then we just sum the semantic contexts and audio features, and this summation result will be sent to the subsequent TFBlock for further modeling. Table 5 shows that CCFM consistently delivers performance improvements over the simple addition fusion approach, regardless of the specific manner of SeCM.

| Fusion Strategies | -15dB | | | -10dB | | | -5dB | | | 0dB | | |
|---|---|---|---|---|---|---|---|---|---|---|---|---|
| | SDR | PESQ | STOI | SDR | PESQ | STOI | SDR | PESQ | STOI | SDR | PESQ | STOI |
| Add (naive) | 5.414 | 1.973 | 0.779 | 8.637 | 2.307 | 0.845 | 11.526 | 2.663 | 0.891 | 14.210 | 3.022 | 0.924 |
| CCFM (ours) | **5.814** | **1.988** | **0.782** | **8.986** | **2.321** | **0.847** | **11.768** | **2.673** | **0.892** | **14.354** | **3.031** | **0.925** |

(a) Evaluation of different contextual information fusion strategies, with $SeCM_V$ to obtain semantic context.

| Fusion Strategies | -15dB | | | -10dB | | | -5dB | | | 0dB | | |
|---|---|---|---|---|---|---|---|---|---|---|---|---|
| | SDR | PESQ | STOI | SDR | PESQ | STOI | SDR | PESQ | STOI | SDR | PESQ | STOI |
| Add (naive) | 6.680 | 2.214 | 0.833 | 9.838 | 2.548 | 0.879 | 12.588 | 2.865 | 0.912 | 15.012 | 3.189 | 0.937 |
| CCFM (ours) | **7.284** | **2.269** | **0.842** | **10.385** | **2.608** | **0.886** | **12.998** | **2.922** | **0.917** | **15.339** | **3.235** | **0.939** |

(b) Evaluation of different contextual information fusion strategies, with $SeCM_{PAV}$ to obtain semantic context.

Table 5: Evaluation of CCFM.

| US | FM | -15dB | | | -10dB | | | -5dB | | | 0dB | | |
|---|---|---|---|---|---|---|---|---|---|---|---|---|---|
| | | SDR | PESQ | STOI | SDR | PESQ | STOI | SDR | PESQ | STOI | SDR | PESQ | STOI |
| ✓ | ✓ | **7.284** | **2.269** | **0.842** | **10.385** | **2.608** | **0.886** | **12.998** | **2.922** | **0.917** | **15.339** | **3.235** | **0.939** |
| ✗ | ✓ | 6.992 | 2.252 | 0.839 | 10.098 | 2.593 | 0.884 | 12.737 | 2.918 | 0.915 | 15.064 | 3.229 | 0.937 |
| ✓ | ✗ | 7.025 | 2.214 | 0.835 | 10.075 | 2.560 | 0.882 | 12.681 | 2.876 | 0.914 | 15.012 | 3.189 | 0.937 |
| ✗ | ✗ | 6.803 | 2.199 | 0.831 | 9.866 | 2.535 | 0.879 | 12.559 | 2.869 | 0.913 | 14.950 | 3.190 | 0.936 |

Table 6: Ablation study of visual frequency for AVSE. US indicates frequency upsample operation; FM indicates frequency modeling module in CCFM.

**Visual Frequency:** Besides evaluating the effect of semantic and signal contexts, this paper also underscores the significance of visual information in recovering audio frequency domain information, an aspect always overlooked in previous studies. Specifically, we performed ablation experiments on the frequency upsampling component of the CCFM and the frequency fusion module. Table 6 illustrates that both the visual frequency upsampling operation and the frequency fusion module significantly improve the performance of the AVSE model, with their combination yielding the best results. This highlights the importance of visual frequency, and suggests that reevaluating visual contributions to frequency characteristics could advance multimodal signal processing.

## 5 CONCLUSION

In this study, we introduce a novel framework, Cross-Modal Contextual Modeling ($CM^2$), inspired by the phonemic restoration phenomenon observed in the human auditory system. This approach enhances AVSE by leveraging two types of contextual information: semantic and signal contexts. Additionally, we incorporate the visual information into the frequency domain, a critical aspect often overlooked in previous research. Our method consistently outperforms existing techniques across all metrics with a wide SNR ranges. Through systematic ablation studies, we have validated the effectiveness of our proposed semantic contexts, signal contexts, and the integration of visual frequency dimensions.

## REPRODUCIBILITY

All data utilized in our study are publicly accessible, and available for use upon application. Our code is implemented using Python 3.9 with Torch version 1.13. All results presented in tables and figures within the paper have been archived. Detailed descriptions of our model are provided in Section 3. We have also detailed the division and processing methodologies for each dataset in Section 4, Appendix C and A

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

## A    Hyperparameter setting

The initial learning rate for our model is established at 0.001, with a batch size of 48. Training is conducted throughout 20 epochs. The StepLR strategy is utilized with step size of 5 and a gamma value of 0.5. The AdamW optimizer is used for training, configured with a weight decay parameter of 0.01 and beta coefficients of (0.9, 0.999).

## B    Evaluation Metrics

We employ three popular metrics for evaluation, including the the Signal-to-Distortion Ratio (SDR) (Raffel et al., 2014), Short-Time Objective Intelligibility (STOI) (Taal et al., 2011), and the Perceptual Evaluation of Speech Quality (PESQ) (Rix et al., 2001). SDR is utilized to evaluate the quality of speech by measuring the ratio of signal power to distortion between the enhanced version and the original clean speech signals. STOI quantifies the intelligibility of a signal on a scale from 0 to 1, with higher values indicating better intelligibility. PESQ assesses the overall perceptual quality of the output signal on a scale from 0.5 to 4.5, where higher values denote superior quality.

## C    Dataset Details

We conducted our AVSE experiments on LRS3+DNS4, GRID+CHiME3, TCD-TIMIT+NTCD-TIMIT, and MEAD+DEMAND datasets. In each dataset group, the first dataset comprises clean audio-visual data, while the second dataset consists of noise samples collected from a variety of settings. For all visual inputs, $SeCM_V$ processes a $112 \times 112$ grayscale image of the face following Wang et al. (2023). Meanwhile, $SeCM_{PV}$ and $SeCM_{PAV}$ receive an $88 \times 88$ grayscale image of the mouth Region of Interest (ROI), in accordance with the configurations of Shi et al. (2022a); Ma et al. (2023); Zhu et al. (2024). In the training phase, the 112x112 facial images and the 88x88 mouth ROI images are randomly cropped from larger 128x128 facial images and 96x96 mouth images, respectively. In the testing phase, a center cropping technique is employed.

**LRS3 + DNS4:** LRS3 contains 438 hours of talking videos from TED and TEDX clips downloaded from YouTube. We evaluate our method on the pretrain subset which contains 407 hours of video. We partitioned this subset into training, validation, and testing sets with a ratio of 8:1:1. We follow Defossez et al. (2020) to obtain the noise signal from the noise subset of the DNS4 dataset. The subset contains approximately 181 hours of noise audio collected from a wide variety of events. During training and evaluation, we utilized these samples as background noise to add noise to the clean speech and construct synthetic noisy audio inputs.

**GRID + CHiME3:** GRID consists of 33 speakers. For our experiments, we follow the general setting Balasubramanian et al. (2023) to designate speakers s2 and s22 as the validation set, speakers s1 and s12 as the unseen unheard test set, and the remaining 29 speakers as the training set. We sample noise from CHiME to corrupt the clean speech. The noise in CHiME is categorized into 4 types: Cafe, Street, Bus, and Pedestrian. The CHiME dataset is divided into training, validation and testing sets with an 8:1:1 ratio.

**TCD-TIMIT + NTCD-TIMIT:** TCD-TIMIT consists of AV speech data from 56 English speakers with an Irish accent. Each utterance is approximately 5 seconds long and sampled at 16kHz. As recommended in Harte & Gillen (2015), we split the dataset into training, validation, and testing sets, with 39 speakers for training, 8 for validation, and 9 for testing. The noisy speech input is derived from the NTCD-TIMIT dataset. This dataset is created by adding six different types of noise to the original speech data from the TCD-TIMIT corpus. The noise types include Living Room, White, Cafe, Car, Bable, and Street, and each noise type is associated with a specific SNR. Similar to the approach in Golmakani et al. (2023), we selected 5 utterances per noise level and noise type for each test speaker to create a test set of 1350 utterances.

**MEAD + DEMAND:** The MEAD dataset consists of recordings from 46 participants, who uttered sentences expressing eight different emotions at three intensity levels under seven camera viewpoints. This dataset is extensively employed in research across various fields, including affective computing, human-computer interaction, and robust AVSE. Following Kang et al. (2022), choose videos that captured frontal views and the highest level (level 3) of emotion intensity for experiment.

For training, we utilized approximately 5 hours of videos from the MEAD dataset. Additionally, 0.7 hours were reserved for validation, and another 0.7 hours were allocated for testing purposes. The DEMAND dataset comprises noise recordings from multiple real-world environments and is extensively used in fields such as speech enhancement and speech recognition.

## D    COMPARISONS RESULTS ON GRID + CHiME3

We compare the proposed $CM^2$ with the SOTA AVSE approaches on GRID datasets. Following Wang et al. (2020b), we utilize the noises from the CHiME3 dataset to synthesize the noisy input audios and perform an evaluation with the test signal-to-noise ratio (SNR) levels of both -5dB and 0dB. As shown in Table 7, $CM^2$ achieves the best performance in both the PESQ improvement (PESQi) and STOI improvement (STOIi) metrics with different test SNR levels.

| Model | -5dB | | 0dB | |
|---|---|---|---|---|
| | STOIi(%) | PESQi | STOIi(%) | PESQi |
| L2L 2018 | 11.14 | 0.54 | 8.86 | 0.62 |
| VSE 2018a | - | 0.45 | - | 0.60 |
| OVA 2020b | - | 0.40 | - | 0.66 |
| VSET 2021 | - | 0.50 | - | 0.75 |
| MHCA-AVCRN 2022 | 13.51 | 0.76 | 11.25 | 0.88 |
| M3Net 2023 | 13.42 | 0.75 | 11.31 | 0.89 |
| DualAVSE 2023 | 15.79 | 0.76 | 13.56 | 0.92 |
| $CM^2$ | **26.25** | **1.01** | **20.50** | **1.21** |

Table 7: Comparison of $CM^2$ with existing AVSE methods on GRID + CHiME3 datasets. '-' denotes that the results are not reported in the original paper.

## E    COMPARISONS RESULTS ON TCD-TIMIT + NTCD-TIMIT

We further evaluate our $CM^2$ model on the TCD-TIMIT dataset, comparing it with SOTA AVSE methods. The reporting metrics in Golmakani et al. (2023) contains SI-SDR (Le Roux et al., 2019), PESQ, and STOI. We report the score improvement as a means of comparison. As illustrated in Table 8, the proposed $CM^2$ achieves the best performance across all metrics at all SNR levels.

| Model | SI-SDRi (dB) | | | | | PESQi | | | | | STOIi(%) | | | | |
|---|---|---|---|---|---|---|---|---|---|---|---|---|---|---|---|
| | -5 | 0 | 5 | 10 | 15 | -5 | 0 | 5 | 10 | 15 | -5 | 0 | 5 | 10 | 15 |
| A-VAE 2021 | 4.34 | 5.12 | 5.93 | 6.07 | 5.76 | 0.16 | 0.19 | 0.20 | 0.21 | 0.05 | 2 | 2 | 4 | 4 | 4 |
| AV-VAE 2021 | 6.15 | 6.86 | 7.38 | 7.22 | 6.52 | 0.24 | 0.27 | 0.29 | 0.28 | 0.08 | 2 | 3 | 4 | 5 | 4 |
| A-DKF 2023 | 5.78 | 6.80 | 7.67 | 8.35 | 7.71 | 0.27 | 0.32 | 0.36 | 0.38 | 0.18 | 2 | 5 | 7 | 9 | 8 |
| AV-DKF 2023 | 9.02 | 9.50 | 10.10 | 9.62 | 8.56 | 0.43 | 0.48 | 0.49 | 0.43 | 0.20 | 5 | 8 | 9 | 10 | 8 |
| DualAVSE2023 | 18.50 | 17.18 | 15.35 | 12.93 | 10.71 | 0.45 | 0.67 | 0.88 | 1.06 | 1.16 | 15 | 15 | 13 | 10 | 6 |
| $CM^2$ | **20.89** | **19.95** | **18.39** | **16.27** | **13.86** | **1.21** | **1.50** | **1.77** | **1.97** | **2.02** | **27** | **26** | **22** | **16** | **11** |

Table 8: Comparison resutls on TCD-TIMIT + NTCD-TIMIT datasets.

## F    COMPARISONS RESULTS ON MEAD + DEMAND

We conduct a comparison between $CM^2$ and the AVSE methods on the MEAD dataset. The results presented in Table 9 demonstrate that our proposed $CM^2$ model outperforms all other methods in terms of all evaluated metrics across various SNR conditions.

| Model | SI-SDRi (dB) | | | | | PESQi | | | | | STOIi(%) | | | | |
|---|---|---|---|---|---|---|---|---|---|---|---|---|---|---|---|
| | -10 | -5 | 0 | 5 | 10 | -10 | -5 | 0 | 5 | 10 | -10 | -5 | 0 | 5 | 10 |
| A-VAE 2018 | 8.91 | 10.33 | 10.52 | 9.81 | 8.14 | 0.03 | 0.27 | 0.35 | 0.38 | 0.31 | 1 | 3 | 4 | 1 | -1 |
| AV-CVAE 2020 | 8.96 | 10.58 | 10.45 | 9.46 | 7.65 | 0.12 | 0.32 | 0.39 | 0.37 | 0.31 | 2 | 4 | 3 | 1 | -2 |
| AV-CVAE-WithHM 2022 | 8.08 | 10.02 | 10.12 | 9.21 | 7.70 | 0.12 | 0.29 | 0.32 | 0.30 | 0.28 | 1 | 2 | 1 | -1 | -3 |
| AV-CVAE-RFF 2022 | 9.62 | 10.72 | 10.68 | 9.70 | 8.00 | 0.22 | 0.45 | 0.46 | 0.43 | 0.35 | 3 | 5 | 5 | 1 | -1 |
| DualAVSE 2023 | 16.06 | 15.21 | 14.09 | 12.98 | 11.27 | 0.35 | 0.54 | 0.74 | 0.92 | 1.01 | 10 | 10 | 8 | 5 | **3** |
| CM$^2$ | **22.02** | **21.07** | **19.22** | **16.90** | **14.13** | **1.26** | **1.57** | **1.68** | **1.60** | **1.33** | **17** | **14** | **9** | **6** | **3** |

Table 9: Comparison resutls on MEAD + DEMAND datasets.

# G    STRUCTURES OF SECM

We further explore the impact of employing different pre-trained models as SeCM. Specifically, we substitute SeCM$_{PAV}$ in CM$^2$ with various pre-trained models known for their strong performance in AVSR tasks, including AVHuBERT, VATLm, and Auto-AVSR. Given that these models were pre-trained exclusively on clean audio-visual paired data, they have learned the share information from audio-visual data. While the audio inputs of AVSE are interfered by noise, the share information would be destroyed thus resulting in performance degradation. To ensure a fair comparison, we also investigate a modified variant of SeCM$_{PAV}$, which similarly uses only the visual modality. The experimental results, as shown in Table 10, reveal several key insights: (1) All pre-trained models used as SeCM significantly improve enhancement performance, validating the importance of semantic contextual information in AVSE. (2) SeCM$_{PV}$ (3) Robust AVHuBERT (SeCM$_{PAV}$) shows marked superiority over other pre-trained models, irrespective of whether the input is visual-only or audio-visual. This advantage is attributed to the inclusion of noise during the training process, which enables Robust AVHuBERT to learn more robust audio-visual representations under noisy conditions.

| Model | A-V | -15dB | | | -10dB | | | -5dB | | | 0dB | | |
|---|---|---|---|---|---|---|---|---|---|---|---|---|---|
| | | SDR | PESQ | STOI | SDR | PESQ | STOI | SDR | PESQ | STOI | SDR | PESQ | STOI |
| SeCM$_{PAV}$ | | | | | | | | | | | | | |
| AVHuBERT | AV | **7.284** | **2.269** | **0.842** | **10.385** | **2.608** | **0.886** | **12.998** | **2.922** | **0.917** | **15.339** | **3.235** | **0.939** |
| AVHuBERT | V | 6.789 | 2.231 | 0.836 | 9.869 | 2.543 | 0.878 | 12.511 | 2.869 | 0.911 | 14.916 | 3.196 | 0.935 |
| SeCM$_{PV}$ | | | | | | | | | | | | | |
| AVHuBERT | V | 6.877 | 2.233 | 0.836 | 9.885 | 2.557 | 0.879 | 12.504 | 2.885 | 0.912 | 14.842 | 3.208 | 0.936 |
| VATLM | V | 6.878 | 2.245 | 0.836 | 10.074 | 2.543 | 0.879 | 12.725 | 2.858 | 0.911 | 15.030 | 3.179 | 0.935 |
| VATLM | AV | 6.305 | 2.084 | 0.799 | 9.799 | 2.466 | 0.868 | 12.561 | 2.831 | 0.909 | 15.008 | 3.170 | 0.935 |
| Auto-AVSR | V | 6.384 | 2.126 | 0.813 | 9.588 | 2.455 | 0.866 | 12.299 | 2.798 | 0.904 | 14.692 | 3.131 | 0.930 |
| SeCM$_V$ | | | | | | | | | | | | | |
| VSR | V | 5.814 | 1.988 | 0.782 | 8.986 | 2.321 | 0.847 | 11.768 | 2.673 | 0.892 | 14.354 | 3.031 | 0.925 |

Table 10: Comparisons of different pre-trained models for SeCM. **A-V** indicate whether the inputs to SeCM is visual modality only or audio-visual modality. These models under SeCM$_{PAV}$ are pre-trained on paired video and noisy audio, while the others under SeCM$_{PV}$ are pre-trained on paired video and clean audio.

# H    STRUCTURE OF CCFM

As illustrated in Figure 3, the integrated output includes the residual connection for semantic context, denoted as $Res_E$. To further explore the role of semantic context in the fusion process, we developed a variant of CCFM that omits $Res_E$, simulating conditions devoid of semantic context. This variant differs from the AOSE Baseline presented in Table 2 and Table 3 in that it still employs the AVSE approach but excludes the direct influence of semantic context. Table 11 demonstrates that the residual connections consistently enhance performance. This indicates that semantic context plays

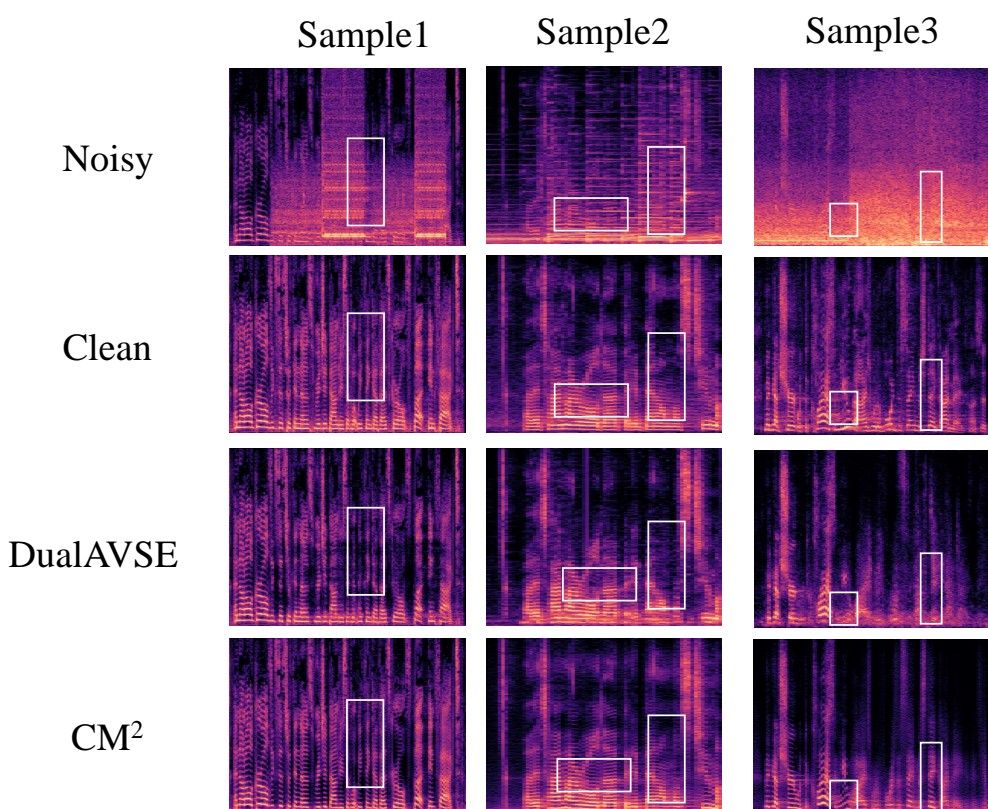

Figure 4: Visualization of the spectrum for three samples.

a role not only during the initial stages of fusion at shallower network layers but also continues to guide performance improvements in deeper network layers where contextual information is fully integrated.

| $Res_E$ | -15dB | | | -10dB | | | -5dB | | | 0dB | | |
|---|---|---|---|---|---|---|---|---|---|---|---|---|
| | SDR | PESQ | STOI | SDR | PESQ | STOI | SDR | PESQ | STOI | SDR | PESQ | STOI |
| ✓ | **7.284** | **2.269** | **0.842** | **10.385** | **2.608** | **0.886** | **12.998** | **2.922** | **0.917** | **15.339** | **3.235** | **0.939** |
| ✗ | 7.033 | 2.220 | 0.835 | 10.036 | 2.560 | 0.880 | 12.697 | 2.900 | 0.914 | 14.947 | 3.203 | 0.936 |

Table 11: Comparison results of different CCFM configurations, $Res_{SeC}$ indicates whether to add semantic context residual connections, as depicted in Figure 3.

## I    VISUALIZATION OF SPECTROGRAMS

Figure 4 displays the spectrograms of three samples, arranged from top to bottom as follows: noisy speech, clean speech, DualAVSE enhancement results, and our $CM^2$ enhancement results. The figure clearly illustrates that our $CM^2$ enhancement results yield a spectrum with clearer and richer details. Particularly in extreme noise conditions, as seen in sample 3, the spectrogram enhanced by DualAVSE appears very blurry, while the $CM^2$ enhancement results are clearer.

