# OpenReview forum: "CM^2: Cross-Modal Contextual Modeling for Audio-Visual Speech Enhancement"
_ICLR.cc/2025/Conference — Submitted to ICLR 2025_

### Official Review · Reviewer_fg1Q · 2024-11-01

**Soundness:** 3
**Presentation:** 2
**Contribution:** 2
**Rating:** 3
**Confidence:** 5

**Summary:**

The paper presents a novel approach called Cross-Modal Contextual Modeling (CM2) for Audio-Visual Speech Enhancement (AVSE). Inspired by the human auditory system's phonemic restoration, CM2 integrates semantic and signal-level contexts across audio and visual modalities to improve speech quality in noisy environments.

**Strengths:**

1. This paper introduces a new approach to audiovisual speech enhancement (AVSE) by integrating semantic and signal context, inspired by phoneme recovery.

2. The semantic context module (SeCM), signal context module (SiCM), and cross-context fusion module (CCFM) are clearly explained in the paper.

3. The authors conduct detailed ablation experiments on the proposed modules, effectively illustrating the framework and its components.

**Weaknesses:**

1. The authors propose a SeCM block that integrates visual and auditory semantic content, but the method does not explain how E is obtained from V, PV, and PAV. The authors should provide a detailed explanation of this process. Additionally, using time-domain information as input increases the complexity of this part of the model.

2. References to BiMamba should include [1,2] because these methods are the first to use bidirectional Mamba in the speech domain, which was not present in the original Mamba paper.

3. The time-frequency alternating module is very common in the speech separation field, and the authors should reference related work. For example, TF-GridNet [3] and RTFSNet [4] use similar time-frequency alternating modules, which are very effective in multimodal speech enhancement.

4. The multimodal speech enhancement methods compared are quite outdated. The authors should compare the latest methods (RTFSNet [4], [5], [6], etc.), as many new methods were proposed in 2024. Moreover, using numerous pre-trained models in the SeCM block results in a very complex model, which might not be optimal compared to current methods, as increased parameters can enhance model generalization. The authors should calculate the parameter count and computational load (MACs) of different models to more comprehensively demonstrate model performance.

5. In Equation 8, the authors did not describe the meaning of the F function, which should be explained there.

6. In line 278, the speech feature P should not be F and T; please correct this.

7. Lines 424 and 425 have insufficient spacing and should be adjusted.

[1] Jiang X, Han C, Mesgarani N. Dual-path mamba: Short and long-term bidirectional selective structured state space models for speech separation[J]. arXiv preprint arXiv:2403.18257, 2024.

[2] Li K, Chen G. Spmamba: State-space model is all you need in speech separation[J]. arXiv preprint arXiv:2404.02063, 2024.

[3] Wang Z Q, Cornell S, Choi S, et al. TF-GridNet: Integrating full-and sub-band modeling for speech separation[J]. IEEE/ACM Transactions on Audio, Speech, and Language Processing, 2023.

[4] Pegg S, Li K, Hu X. RTFS-Net: Recurrent time-frequency modelling for efficient audio-visual speech separation[J]. arXiv preprint arXiv:2309.17189, 2023.

[5] Tiwari U, Gogate M, Dashtipour K, et al. Real-Time Audio Visual Speech Enhancement: Integrating Visual Cues for Improved Performance[C]//Proc. AVSEC 2024. 2024: 38-42.

[6] Gogate M, Dashtipour K, Hussain A. A Lightweight Real-time Audio-Visual Speech Enhancement Framework[C]//Proc. AVSEC 2024. 2024: 19-23.

**Questions:**

Please refer to above.

---

### Official Review · Reviewer_s61k · 2024-11-02

**Soundness:** 3
**Presentation:** 3
**Contribution:** 3
**Rating:** 5
**Confidence:** 5

**Summary:**

This paper introduces a novel framework, called Cross-Modal Contextual Modeling ($CM^2$), for developing audio-visual speech enhancement technology. The $CM^2$ uses two types of contextual information: semantic and signal contexts. Semantic-level context enables the model to infer missing or corrupted content, leveraging semantic consistency across segments, whereas signal-level context further explores coherence within the signals developed from the semantic consistency. The $CM^2$ also incorporates visual information into the frequency domain and verifies that visual information plays a critical role in enhancing noisy speech.

**Strengths:**

- This paper is well-written and easy to understand.
- The authors clearly pointed out that most existing approaches focus solely on fusion in the temporal domain and overlook the potential correlations between the frequency dimensions of the visual and audio modalities.
- The authors conducted comprehensive experiments on ablation studies showing that the proposed $MC^2$ outperforms the previous AVSE methods.

**Weaknesses:**

- The authors underscore the significance of visual information in recovering audio frequency domain information with the ablation study in Table 6. However, my concern is that the performance improvements are mainly due to the pre-trained visual encoder like AV-HuBERT, not the proposed CCFM (also in Table 10). I think it is better to provide more analysis on how the proposed CCFM actually boosts up speech enhancement performances without the largely pre-trained visual encoder. I also suggest the authors provide visualizations of the intermediate features produced by CCFM.

- While the task is audio-visual speech enhancement, the authors have not provided a demo video showing how clear and intelligible the output speech samples are. The desirable demo could be side-by-side comparisons with baseline methods. Furthermore, there is no human subjective MOS performance verifying that the enhanced output samples are actually better than those from the previous literature. I would suggest gathering a certain amount of participants to validate the enhanced speech sample by evaluating naturalness, intelligibility, etc.

- The authors only showed three different metrics while other previous papers have more. I encourage the authors to provide more quantitative performance metrics like MCD which is Mean Cepstral Distance measuring the difference between the spectral features of the synthesized speech and the target speech and ViSQOL which is Virtual Speech Quality Objective Listener to verify the proposed $MC^2$. I think MCD is important because this paper specifically underlines the importance of the frequency-domain characteristics of speech with visual appearance.

**Questions:**

### Additional Comments
- There are missing references like [1,2] that are well-known in speech enhancement tasks. Also, I would recommend the authors compare the proposed architecture with LA-VocE [1] since it's one of the recent state-of-the-art AVSE papers.
- Is there a reason that the authors use ISTFT not vocoder when converting the mel-spectrogram into the actual audio waveform?
- Besides the quantitative comparison, I am curious about the comparisons of inference times and numbers of parameters of the proposed model and other methods. Are those comparable?
- line 269: $CM_2$ -> $CM^2$?
- Please increase the line space between lines 425 - 426 for better readability.

[1] Mira, Rodrigo, et al. "LA-VocE: Low-SNR audio-visual speech enhancement using neural vocoders." ICASSP 2023-2023 IEEE International Conference on Acoustics, Speech and Signal Processing (ICASSP). IEEE, 2023.
[2] Tan, Ke, and DeLiang Wang. "Learning complex spectral mapping with gated convolutional recurrent networks for monaural speech enhancement." IEEE/ACM Transactions on Audio, Speech, and Language Processing 28 (2019): 380-390.

---

### Official Review · Reviewer_6pDV · 2024-11-02

**Soundness:** 2
**Presentation:** 2
**Contribution:** 2
**Rating:** 3
**Confidence:** 3

**Summary:**

This paper presents a framework called Cross-Modal Contextual Modeling (CM2) to improve Audio-Visual Speech Enhancement (AVSE). CM2 integrates two types of contextual information—semantic and signal context—across audio and visual data to enhance speech quality in noisy environments. The semantic context helps the model infer missing or corrupted speech by maintaining consistency across segments, while the signal context leverages coherence within signal frames. Additionally, the approach emphasizes the importance of visual features, such as the speaker's facial cues, which can correlate with audio frequency characteristics, thus aiding the enhancement process.

The model uses three main components to build and integrate these contexts: a Semantic Context Module (SeCM) for initial contextual extraction, a Signal Context Module (SiCM) for signal-level context from noisy inputs, and a Cross-Context Fusion Module (CCFM) to combine these contexts. This architecture allows for detailed context fusion across different modalities, effectively improving speech clarity, especially in challenging low signal-to-noise ratios. Experimental results show that CM2 outperforms other state-of-the-art models, demonstrating substantial gains in metrics related to speech quality and intelligibility.

Methodology:
 - SeCM: Processes Video and Audio in Time Domain audio stream, proposes three solutions (V, PV, PAV) from other papers, and test to find the best approach.
 - Audio Encoder: Processes audio in TFM Domain, uses 4 stacked convolutions.
 - SiCM: uses a bidirectional Mamba approach to process along different dimensions and time directions.
 - Time-Frequency Upsampler: two transpose convolutions with BN and prelu. used to take the frequency dim from 1 to match the frequency dim of the other audio route.
- CSBlock: split across the channels and concatenate cross-modality information
- AF Block - Generate attention map by applying linear to P, use two SiMC on P^c and E^c, then multiply the mask by both features and they are added together. A 2d conv binds the information together.

**Strengths:**

The introduction and literature view show a strong and detailed narrative of their task, their goals and their contributions. It is well researched and has a comprehensive view of current methods, contextualising their methodology and results. Their methods section is long and detailed, with many interesting approaches to the problems that they face. They combine a series of current techniques, such as Mamba and individually processing the time and frequency dimensions, and then applying a global operation down both dimensions. They use a discriminator to add another loss signal to the training process, and in their experiments they cover a range of datasets and evaluation metrics, strengthening their claims. This show good scientific rigour.

**Weaknesses:**

The weaknesses mainly come from technical writing problems and lack of clarity.

 - SiCM is defined, but it is not clear how it is used. This section appears to define the equations twice, which is unnecessary. Its inputs are not defined - what is $Q_{in}$ and how does it connect with the rest of the model? Use of mamba is interesting, bidirectional seems unusual - and they do not have experiments to back up this choice.
 - CM_2:
	- Line 278 - should the notation be $B F_x \times T_x \times C$ instead?
	- Line 283 "diverging" is repeated.
	- Line 301 the $F_e$ and $F_p$ are not clearly defined (1d convolutions?).
	- On equastions 6 and 7 add some spaces after the "..." and after the "," otherwise its too hard to read
	- Line 302: it should be "contains" not "contain"
	- 3.5 the use of the SiCM notation is a little confusing. Earlier, it refers to a specific set of Mamba-based operations, but here it is for the entire frequency processing and time processing modules. Please keep the nomenclature consistent.
 - The Discriminator is mentioned but undefined. Papers need to be reproducible.
 - The methods they compared to in the results tables are quite old. Would be interesting to compare with modern AVSS models by setting speaker 2 = noise.

Finally, while this is not a strict requirement, open sourcing the code after the paper's release would help understand the methods better, as the methods are quite convoluted.

**Questions:**

AF Block:
 - Equation 11 is an interesting operation. I would expect something like $MI + (1-M)I$ if $M$ were a Mask. Could you provide some justification/insight into this operation, such as motivation/related work?
 - "Attention" usually refers to a specific set of operations. To make this block resemble attention, you could apply two SiCMs to $P^c$ (to make a $K$ and $V$), then one SiCM to $E^c$ (to get $Q$), and then create an attention map by applying cross attention with $Q$, $K$ and $V$. Of course, this would be quadratic in $T_x$, so this may be computationally prehibative. Would it be possible to explore alternative operations?

Experiments:
 - Would it be possible to add Si-SNR(i) and SDR(i) metrics to the results?

**Details Of Ethics Concerns:**

- Figure 2 looks very similar to RTFS-Net (https://arxiv.org/abs/2309.17189). There is some overlap between the methods introduced in this paper, and the contributions detailed in RTFS-Net and TF-GridNet (https://arxiv.org/abs/2209.03952). While the finer details differ, the overall approach is very similar. Please add citations.
 - CSBlock: This method of switching the dimensions was first introduced by DPRNN in 2019. Other's work should be properly sited and referenced. This method has been used every single year since 2019, such as by DPTNet in 2020, TF-Gridnet in 2022, and RTFS-Net in 2023.

---

> ### Author Response · Authors · 2024-11-18
> **Response to Reviewer 6pDV for Research Integrity Issues（1/3）**
>
> Dear Reviewer 6pDV,
>
> Thank you for your recognition, suggestions, and evaluation of our work. Regarding the research integrity issues you raised, we must clarify that the contributions we claimed are entirely different from those in the mentioned works like RTFS-Net.
>
> Firstly, you mentioned that our Figure 2 looks similar to RTFS-Net and there is a mount of overlap between the contributions of our paper and the contributions in RTFS-Net and TF-Gridnet. We believe this is a misunderstanding. In AVSE / AVSS, the pipeline of “audio / visual encoders → audio-visual fusion module → (Time-Frequency) enhancer/separator → decoder” is quite common, as seen in OVA in 2020 (https://ieeexplore.ieee.org/document/9053033), Muse in 2021 (https://arxiv.org/abs/2010.07775), and VisualVoice in 2021 (https://arxiv.org/abs/2101.03149). This commonality in the overall pipeline frameworks may cause superficial similarities between different works, especially when using relatively generalized diagrams for overview representation. However, this process itself is common, and also not a contribution we claimed. Considering the generic nature of this pipeline, it is improper to deem our work as potentially plagiaristic which is such a hurtful term. Moreover, the issues addressed by RTFS-Net and TF-Gridnet are entirely different from our paper (speech separation aims to separate different speakers, while speech enhancement aims to enhance target speech from noisy input). Besides this point:
> 1. The first contribution we claimed is that we propose a new perspective to solve AVSE. This perspective draws inspiration from cognitive studies on human auditory processing in noisy environments. Drawing inspiration from how humans use semantic insights to compensate for noise-affected segments, we align this approach with the objective of AVSE—to enhance the output speech signals. Based on this general idea, we extract semantic-level information at first, which then guides the subsequent modeling of speech at the signal level so as to lead to an enhanced, high-quality speech signal at the end. This general idea has not been previously proposed. RTFS-Net's main contribution is a lightweight model that performs AVSS in the time-frequency domain with minimal resource consumption, achieving impressive performance. After carefully reviewing the RTFS-Net，we acknowledge it as an important and influential work in the field of speech separation. We are also pleased to include RTFS-Net and other significant speech separation studies mentioned in your comment. However, we must clarify that our contributions are entirely distinct from these works.
> 2. Our second contribution specifically emphasizes the importance of visual information for modeling the frequency characteristics of audio information—an aspect previously neglected in works like RTFS-Net and other AVSE/AVSS approaches. Earlier time-frequency domain AVSE/AVSS methods, including RTFS-Net, focused solely on aligning audio-visual features in the time dimension. They typically repeated temporal visual features along the audio frequency dimension merely to make the visual feature’s dimension equal to audio feature’s. In contrast, we noticed the role of visual information for recovering audio signals and intentionally introduced simple learnable upsampling of the audio-visual semantic context along the frequency dimension. This simple learnable manner aims to  maximize the influence of visual data on every frequency bins of the audio spectrum.
> 3. Finally, before we discuss the specific module comparisons, it is important to note that our development process was guided by a new perspective and particular objectives. Each module in our system is introduced with unique objectives, distinct from RTFS-Net or any other systems.

---

> ### Author Response · Authors · 2024-11-18
> **Response to Reviewer 6pDV for Research Integrity Issues（2/3）**
>
> a) Firstly, while both RTFS-Net and our method utilize pre-trained models to extract visual features, the ways we use them, the methods by which we obtain these features, and our objectives all vary substantially. RTFS-Net hopes to extract target speaker information from visual lip movements with the pre-trained lipreading network. Conversely, our approach integrates the multi-modal semantic information from both noisy audio and visual inputs by the self-supervised audio-visual speech model (AV-HuBERT), which is used to guide and enhance the subsequent construction of signal-level context. Utilizing AVHuBERT or similar models to encode visual information is common in many AVSE/AVSS applications. For instance, AV-Gen in 2023 (https://arxiv.org/abs/2306.01432) employs AVHuBERT to derive visual embeddings, where the visual features of each layer in AVHUBERT would be weighted to sum to inform an audio-based Score Modal  for AVSE; SSL-AVSE in 2023 (https://arxiv.org/abs/2307.07748) leverages AVHuBERT as a front-end encoder for audio-visual feature encoding, where the audio-visual features in each layer are weighted to sum to feed a decoder for AVSE. These implementations are distinctly different from ours in both purpose and the subsequent manner to use the obtained features.
>
>    b) Secondly, as detailed in Section 3.2, our audio encoder consists of a validated structure comprising four convolutional blocks. This configuration was determined through our experiments and is independent of the audio encoder used in RTFS-Net.
>
>    c) Thirdly, concerning the modal fusion module, we highlight three key differences between our CM$^2$ module and RTFS-Net:
>
>       i) Overall, our CCFM is not a typical modal fusion module. CCFM is a context fusion module, as detailed in Sections 1 and 3.4, differing significantly from RTFS-Net’s CAF Block in both target and specific design. CCFM focus on using semantic context to guide signal context modeling. Initially, it upsamples the multi-modal semantic context, and then combines it with the signal context at the channel level. This is followed by a fine-grained fusion process, implemented in an attention-like manner, to enhance integration. This approach is totally different from RTFS-Net’s method of Gated Fusion for audio-visual features, as evidenced by the distinct methodologies depicted in our Figure 3 compared to RTFS-Net Figure 2.
>
>       ii) Moreover, we emphasize the impact of speaker’s visual appearance on the audio frequency domain. As mentioned in the previous point 2, most TF-domain AVSS / AVSE methods, including RTFS-Net, focus only on fusing audio-visual features with the same dimension. These methods typically upscale the visual features in temporal domain and then repeat them across the frequency dimension, with the target of just obtaining the same dimension as audio features in the TF-domain. In contrast, we intentionally introduce the learnable upsample to the audio-visual semantic context features in both time and frequency dimensions. Then, in addition to the fusion module on the time dimension, a frequency dimension fusion module is applied for deeper integration of the two types of context. Table 6 demonstrates the effectiveness of this design.
>
>       iii) The channel operations in our CCFM's Channel Swapping Block (CSBlock) differ significantly from those in RTFS-Net. RTFS-Net's operations are concentrated on the real and imaginary components of the audio spectrum, assigning the first half channel dimensions of audio features to the real part and the second half to the imaginary, then employing a complex multiplication-like method for spectral output. However, as detailed in Section 3.4, our CSBlock operates under our premise that different channel information highlights different aspects, and redundant information in a channel of one modality might be complementary for the other modality. Hence, we swap half the channels between two modalities to facilitate preliminary fusion. This method is distinct from RTFS-Net.
>
>    d) In discussing Time-Frequency (TF) domain modeling, it is important to note that modeling time and frequency separately is a standard practice in SE and SS tasks, as seen in TF-Gridnet (2022-9-8 released, ICASSP 2023) and RTFS-Net(2023-9-29 released, ICLR 2024), and earlier in CMGAN (2022-3-28 released, Interspeech 2022, which is already cited in our paper) (https://arxiv.org/abs/2203.15149). Moreover, the TF domain modeling itself is not among our claimed contributions. Instead, we emphasize the significant role of the visual modality in restoring audio frequency domain characteristics, as detailed previously in point 2.
>
>    e) Our decoders are implemented based on CMGAN and are totally unrelated to RTFS-Net or TF-Gridnet.
>
>    f) We implemented a discriminator inspired by CMGAN and Metric GAN, transforming the non-differentiable metric PESQ into a training target. Neither RTFS-Net nor TF-Gridnet employs a similar approach.

---

> ### Author Response · Authors · 2024-11-18
> **Response to Reviewer 6pDV for Research Integrity Issues（3/3）**
>
> For the second question, you mentioned that the operation in the CSBlock was first introduced in DPRNN (2019) and subsequently used in DPTNet (2020), TF-Gridnet (2022), and RTFS-Net (2023). We have conducted extensive research and identified a significant misunderstanding on the part of the reviewer. As detailed in Section 3.4 and previously clarified, our CSBlock is specifically responsible for the preliminary fusion of two different levels of the context. This contrasts sharply with the single-modality models DPRNN, DPTNet, and TF-Gridnet mentioned by the reviewer. This fusion is based on our channel redundancy premise among different modalities as explained previously, and we did not identified any similar ideas or operations in these methods. The only possible similarity to RTFS-Net seems to be the point that both methods involve operations on channel dimensions; however, as we have already clarified as above, their approach to channel operations is different from ours.
> Regarding the misunderstanding, we suspect that the reviewer may have confused our channel swapping operation with the segmentation operation introduced in DPRNN. This segmentation technique divides long input sequences into shorter chunks, which are then concatenated to form a 3-D tensor. Subsequently, two separate RNNs process these sequences along two dimensions of the tensor, a method depicted in DPRNN Figure 1. Similar approaches have been adopted in subsequent mentioned works like DPTNet, TF-Gridnet, and RTFS-Net. However, our CM$^2$ module does not utilize this segmentation method.
>
> Finally, we appreciate your valuable feedback on our work. We hope the explanations provided could resolve any misunderstandings. Should you have any further inquiries or require additional clarification, please do not hesitate to contact us. We are fully willing to address any further concerns you might have and look forward to your response. Additionally, we kindly request you revising the wording of the original review to remove some hurtful wordings such as "plagiarism".

---

> > ### Comment · Reviewer_6pDV · 2024-11-25
> >
> > The points raised are valid, but do not address the problem.
> >
> > To be clear, I think this work differs significantly from previous works and contributions. The introduced techniques are interesting, and I believe they will be helpful to future researchers. However, many of the techniques used have been introduced previously and are uncited in the manuscript.
> >
> > In particular, the claim of ownership of the method of switching dimensions introduced in DPRNN. Specifically, in line 292, the authors state, "we introduce a Channel Swapping Block (CSBlock)." However, this approach is not the original work of $CM^2$, as it can be observed in the DPRNN paper (https://arxiv.org/abs/1910.06379) and source code at lines 312 and 319 (https://github.com/asteroid-team/asteroid/blob/154c52f6e15de9e42213b9997aa1a0ad8b0d453b/asteroid/masknn/recurrent.py#L319). This method has been employed consistently over the past five years, which was my primary reason for raising the ethics flag.
> >
> > The subsequent channel-swapping operation is also reminiscent of equations 23 and 24 of RTFS-Net, and the overall pipeline of $CM^2$ aligns closely with other works such as RTFS-Net, TF-GridNet, CTCNet and others. Another reviewer has also noted the failure to cite other significant papers in the field, which lead me to mistakenly assign the work in this paper as original contributions introduced by $CM^2$, instead of by the original authors. This omission is concerning to me, as I believe this style of writing could easily mislead others in a similar way.
> >
> > Sources should be properly cited and acknowledged. Despite this paper’s similarities to other work, with proper citing and referencing it would not be cause for concern.

---

> > > ### Author Response · Authors · 2024-12-02
> > > **Response to Reviewer 6pDV (1/2)**
> > >
> > > Dear Reviewer 6pDV,
> > >
> > > 1. **Regarding your statement that ‘Specifically, in line 292, the authors state, "we introduce a Channel Swapping Block (CSBlock)."’**
> > >
> > > Would you mind kindly quoting the full sentence, including the preceding and following parts in our paper? The complete sentence in this part reads:
> > >    > 'Given the distinct emphasis of channel information within the two modalities, channels that are redundant in one modality can provide complementary benefits to the other. Based on this idea, we introduce a Channel Swapping Block (CSBlock) to preliminarily merge information across the modalities at the channel level for later fine-grained fusion on time or frequency dimensions.'
> > >
> > > The sentence before states the rationale for introducing the CSBlock, emphasizing our assumption about how the two modalities interact in the channel dimension. The second half of the sentence explains the purpose behind our introduction of the block. The core operation of DPRNN is to solve the ineffectiveness of conventional RNNs in tackling long sequences, by splitting long sequences into smaller chunks and then interleaving two RNNs, an intra-chunk RNN and an inter-chunk RNN, for local and global modeling respectively. As we have emphasized in both the original manuscript and our previous responses, both the motivation and design of our CSBlock are completely different from the operation in DPRNN. Neither previous AVSE nor AVSS methods employ the same approach.
> > >
> > > **Code-level:** In the view of specific implementation, we have reviewed the DPRNN code according to the provided link and provide a line-wise comparison between the relevant DPRNN code and our implementation (Formulas 6 and 7). We truly cannot see any relationship between the DPRNN operation and our CSBlock.
> > >
> > >  **DPRNN Source Code:**
> > > ```py
> > > 309 B, N, K, L = x.size()
> > > 310 output = x  # for skip connection
> > > 311 # Intra-chunk processing
> > > 312 x = x.transpose(1, -1).reshape(B * L, K, N)
> > > 313 x = self.intra_RNN(x)
> > > 314 x = self.intra_linear(x)
> > > 315 x = x.reshape(B, L, K, N).transpose(1, -1)
> > > 316 x = self.intra_norm(x)
> > > 317 output = output + x
> > > 318 # Inter-chunk processing
> > > 319 x = output.transpose(1, 2).transpose(2, -1).reshape(B * K, L, N)
> > > 320 x = self.inter_RNN(x)
> > > 321 x = self.inter_linear(x)
> > > 322 x = x.reshape(B, K, L, N).transpose(1, -1).transpose(2, -1).contiguous()
> > > 323 x = self.inter_norm(x)
> > > ```
> > > **Our Code:**
> > > ```python
> > > 161 audio_first_half = audio[:,//2,...]
> > > 162 video_first_half = video[:,//2,...]
> > > 163 audio_swap = torch.cat([audio_first_half, video[:,C//2:,...]], dim=1)
> > > 164 video_swap = torch.cat([video_first_half, audio[:,C//2:,...]], dim=1)
> > > ```
> > >
> > >    **Equation:** The formulas (23) and (24) from RTFS-Net are to split the audio feature channels into real and imaginary parts, and then calculate the target speech using a form of complex multiplication. While both RTFS-Net method and ours involve splitting the feature channels into two parts, the key point lies in how these parts are used for what purpose. Our goal is to exchange the different channel information between two different modalities based on the redundancy and complementary assumption described above, which is fundamentally different in both usage and purpose compared to RTFS-Net. It is not reasonable to assume that they are similar just because both involve splitting the data into halves.
> > >
> > > 2. **Regarding the overall pipeline of CM$^2$:**
> > >
> > > We have already partially addressed this issue in our previous responses:
> > >
> > > > 'In AVSE / AVSS, the pipeline of “audio / visual encoders → audio-visual fusion module → (Time-Frequency) enhancer/separator → decoder” is quite common, as seen in OVA in 2020 (https://ieeexplore.ieee.org/document/9053033), Muse in 2021 (https://arxiv.org/abs/2010.07775), and VisualVoice in 2021 (https://arxiv.org/abs/2101.03149). This commonality in the overall pipeline frameworks may cause superficial similarities between different works, especially when using relatively generalized diagrams for overview representation. This commonality in the overall pipeline frameworks may cause superficial similarities between different works, especially when using relatively generalized diagrams for overview representation. However, this process itself is common, and also not a contribution we claimed.'
> > >
> > > The works we cited above have already been referenced in our manuscript. Additionally, all the mentioned works in the reviewer’s comment, RTFS-Net, TF-GridNet, and CTCNet share a similar overall pipeline to ours, which means all these works themselves share a similar pipeline. Then does that mean all these works have ethic issues? Clearly, no, because the overall pipeline is not the main claimed contribution of any of these works, nor is it our claimed contribution.

---

> > > > ### Author Response · Authors · 2024-12-02
> > > > **Response to Reviewer 6pDV (2/2)**
> > > >
> > > > 3. **Regarding the citations of prior works.**
> > > >
> > > > Firstly, we must clarify that we are more than willing to include relevant references in the subsequent revision of our paper. However, it is unreasonable and unfair to escalate this to ethical issues simply because we did not cite the mentioned Speech Separation studies. All the mentioned papers in your previous comment belong to the field of Speech Separation (SS), whereas our paper focused on Speech Enhancement (SE). These two fields address fundamentally different problems. Generally, papers on SE do not frequently cite SS literature, and vice versa. For instance, in our literature review, RTFS-Net from the speech separation field cited 13 SS papers but only 2 SE papers. Similarly, CMGAN from the speech enhancement field cited 6 SS papers and 20 SE papers. Our own work, CM², cited 7 SS papers and 32 SE papers. Suggesting that our failure to cite speech separation papers indicates ethic issue is unjustifiable.
> > > >
> > > > Regarding the works mentioned by other reviewers, the main reasons are twofold: to enhance our work by adding additional performance comparisons and to enrich it by including more references. The reasons are not that our ideas are similar to or replicate existing studies. Specifically, the two works mentioned by Reviewer s61k, namely LA-VocE and GCRN, were not previously cited because they represent a different methodology and are not closely related to our method. Nevertheless, we are happy to adopt the reviewer's suggestion to include a performance comparison with these two works. Regarding the comments from Reviewer fg1Q about Dual-path Mamba, SPMamba, TF-Gridnet, and RTFS-Net, we did not previously cite these as they primarily focus on speech separation. However, we are more than willing to reference them to enrich our paper. As for the AVDPRNN (https://www.isca-archive.org/avsec_2024/gogate24_avsec.pdf) and A-V Demucs (https://www.isca-archive.org/avsec_2024/tiwari24_avsec.pdf) mentioned by Reviewer fg1Q, these papers were published just before the ICLR 2024 submission deadline (September 1, 2024). It is generally understandable that such recently published works were not cited. We are grateful for all the reviewers' suggestions regarding additional references. These will certainly help us improve our work. As for the time-frequency alternating modeling methods mentioned by Reviewer fg1Q, we have already cited the CMGAN which is an earlier work that adopted this approach in the field of Speech Enhancement.
> > > >
> > > > We hope the above explanations address your concerns. We once again kindly request you revising the hurtful wordings in the previous review comments. Looking forward to your response.

---

### Official Review · Reviewer_NSCs · 2024-11-11

**Soundness:** 2
**Presentation:** 2
**Contribution:** 2
**Rating:** 5
**Confidence:** 3

**Summary:**

The paper proposed a deep learning model called Cross-Modal Contextual Modeling (CM^2), which utilizes both audio and visual cues to enhance speech quality in noisy environments. CM^2 combines two types of contextual information: semantic context and signal context. Experimental results show superior model performance over existing works.

**Strengths:**

The motivation is sound and clear
The paper reports SOTA performance on all metrics including SDR, PESQ, and STOI.

**Weaknesses:**

The idea of extracting and integrating semantic context, signal context, and visual frequency (as shown in figure 1) is quite interesting, but the connections between these components and the proposed model modules/ architecture are not so strong/ clear. Semantic context is extracted from a pretrained model (visual or audio-visual model, e.g. AVHuBERT), while signal context is simply like a fusion module of noisy audio input and the semantic features.
The proposed model architecture consists of many components including SeCM, CCFM, SiCM, and a pretrained model like AVHuBERT; it was trained with a loss function of magnitude spectrogram loss, complex spectrogram loss, and adversarial loss. Thus, it is hard to highlight the importance of each component.
Experimental results mainly focus on model performance on speech enhancement with metrics of SDR, PESQ, and STOI. However,  analysis of the model resource consumption including memory and computational cost (at training and inference) could be beneficial.

**Questions:**

About evaluation, how do you make comparision with previous works under different signal-to-noise ratio conditions as I don't see this type of evaluation on most of the previous works?

---

### Meta-Review · Area_Chair_n7sr · 2024-12-19

**Metareview:**

This paper presents Cross-Modal Contextual Modeling (CM^2) to improve Audio-Visual Speech Enhancement (AVSE). The idea is to use visual information to de-noise or otherwise improve the quality of audible speech. CM^2 integrates two types of contextual information—semantic and signal context. The semantic context helps the model infer missing or corrupted speech by maintaining consistency across segments, while the signal context leverages coherence within signal frames. The approach exploits the correlation between visual features, such as speaker's facial cues, and audio frequency characteristics, to aid the enhancement process.

The model consists of three main components: a Semantic Context Module (SeCM) for initial contextual extraction, a Signal Context Module (SiCM) for signal-level context from noisy inputs, and a Cross-Context Fusion Module (CCFM) to combine these contexts. This architecture allows for detailed context fusion across different modalities, to improve speech clarity. Experimental results show that CM^2 outperforms other state-of-the-art models such as RTFD-Net, demonstrating substantial gains in metrics related to speech quality and intelligibility (SDR, PESQ, and STOI).

Strengths
- the paper discusses a relevant problem and the addition of AV-Hubert as a feature extractor is new
- the paper is generally well written and easy to understand

Weaknesses
- Authors do not provide human validations of the improved intelligibility and speech quality (only automatic metrics)
- Stronger & more recent baselines such as RTFSNet are missing (from the paper).

Addressing the two weaknesses is a recommended step before acceptance for publication can be recommended.

**Additional Comments On Reviewer Discussion:**

One reviewer initially considered similarities between the present paper and other published work a potential ethics violation, since some references were not clearly marked. This issue could be resolved as a mis-understanding, and further discussion focused on actual similarities and their impact. Authors however did not actually include comparisons to more recent work in their manuscript.

---

### Decision · Program_Chairs · 2025-01-22

Reject